# 30-year trends in major cardiovascular risk factors in the Czech population, Czech MONICA and Czech post-MONICA, 1985 – 2016/17

Renata Cífková[1,2]*, Jan Bruthans[1], Peter Wohlfahrt[1], Alena Krajčoviechová[1], Pavel Šulc[1], Marie Jozífová[1], Lenka Eremiášová[2], Jan Pudil[2], Aleš Linhart[2], Jiří Widimský Jr[3], Jan Filipovský[4], Otto Mayer Jr[4], Zdenka Škodová[5], Rudolf Poledne[6], Petr Stávek[6], Věra Lánská[7]

1 Center for Cardiovascular Prevention, First Faculty of Medicine and Thomayer Hospital, Charles University in Prague, Prague, Czech Republic, 2 Department of Medicine II, Charles University in Prague, First Faculty of Medicine, Prague, Czech Republic, 3 Department of Medicine III, Charles University in Prague, First Faculty of Medicine, Prague, Czech Republic, 4 Department of Medicine II, Faculty of Medicine, Charles University, Pilsen, Czech Republic, 5 Department of Preventive Cardiology, Institute for Clinical and Experimental Medicine, Prague, Czech Republic, 6 Atherosclerosis Research Laboratory, Institute for Clinical and Experimental Medicine, Prague, Czech Republic, 7 Medical Statistics Unit, Institute for Clinical and Experimental Medicine, Prague, Czech Republic

* renata.cifkova@ftn.cz

**Data Availability Statement:** The data underlying the results presented in the study are available from http://www.ftn.cz/data-monica-1117/

## Abstract

### Background

Compared with Western Europe, the decline in cardiovascular (CV) mortality has been delayed in former communist countries in Europe, including the Czech Republic. We have assessed longitudinal trends in major CV risk factors in the Czech Republic from 1985 to 2016/17, covering the transition from the totalitarian regime to democracy.

### Methods

There were 7 independent cross-sectional surveys for major CV risk factors conducted in the Czech Republic in the same 6 country districts within the WHO MONICA Project (1985, 1988, 1992) and the Czech post-MONICA study (1997/98, 2000/01, 2007/08 and 2016/2017), including a total of 7,606 males and 8,050 females. The population samples were randomly selected (1%, aged 25–64 years).

### Results

Over the period of 31/32 years, there was a significant decrease in the prevalence of smoking in males (from 45.0% to 23.9%; *p* < 0.001) and no change in females. BMI increased only in males. Systolic and diastolic blood pressure decreased significantly in both genders, while the prevalence of hypertension declined only in females. Awareness of hypertension, the proportion of individuals treated by antihypertensive drugs and consequently hypertension control improved in both genders. A substantial decrease in total cholesterol was seen

**Funding:** The study was supported by grant No. 15-27109A provided by the Ministry of Health of the Czech Republic.

**Competing interests:** The authors have declared that no competing interests exist.

in both sexes (males: from 6.21 ± 1.29 to 5.30 ± 1.05 mmol/L; $p < 0.001$; females: from 6.18 ± 1.26 to 5.31 ± 1.00 mmol/L; $p < 0.001$).

## Conclusions

The significant improvement in most CV risk factors between 1985 and 2016/17 substantially contributed to the remarkable decrease in CV mortality in the Czech Republic.

## Introduction

Cardiovascular disease (CVD) remains the most common cause of death in the majority of countries worldwide [1]. During the past 40 years, there has been an increasing interest in geographical variations and time trends of total and cardiovascular (CV) mortality in Europe [2–6]. Death rates from coronary heart disease (CHD) and stroke have been generally higher in Central and Eastern Europe, in countries ruled by the totalitarian regimes for four decades after World War II, compared to Northern, Southern, and Western Europe, enjoying political freedom and free-market economy within the same period [7].

A decline in CVD mortality has been reported in most European countries, including the former communist countries in Central and Eastern Europe, however, the decline in CVD mortality in this part of Europe started substantially later. An analysis by Hartley et al. [5] has claimed that mortality started to decline only 5 years after the political changes occurred. The consequence has been a growing disparity in CVD mortality between Western and Eastern Europe.

Despite the significant decline in CVD mortality in the Czech Republic documented since 1985 (by more than 60% in both genders), the CVD mortality rates have remained high (with the latest available data for 2017: age-standardized mortality in males 334.2/100,000 population, in females 218.0/100,000 population) [8]. Data from the Czech MONICA and the Czech post-MONICA studies were used to develop the validated IMPACT mortality model to explain the decline in CHD mortality in the Czech Republic between 1985 and 2007 [9]. More than half (52%) of the substantial fall in CHD mortality was attributable to the reduction in major CV risk factors. The largest reduction in CHD death (39.5%) was explained by a substantial reduction in total cholesterol (from 6.1 mmol/l in 1985 to 5.1 mmol/l in 2007), predominantly due to lifestyle and dietary changes. Improvement in treatments accounted for approx. 43% of the CHD mortality decrease.

The aim of this analysis was to further assess the longitudinal trends in major CV risk factors in a representative Czech population sample from 1985 to 2016/2017. The study period included also the transition from the totalitarian regime to democracy, associated with socio-economic changes. In addition to our previous analysis [10], the results of the latest survey (2016/17) have been provided together with the longitudinal analysis by 10-year age groups for each of the risk factors.

## Methods

### Study population

There were seven independent cross-sectional surveys for major CV risk factors conducted in the Czech Republic in the same 6 country districts between 1985 and 2016/17. The first three surveys (1985, 1988, and 1992) were carried out within the WHO MONICA Project [11].

The study population was always randomly selected as a 1% population sample within each district, stratified by age and sex, within an age range of 25 to 64 years, either from the National Population Register (1985, 1988, and 1992; the Czech MONICA Study) or from the General Health Insurance Company registry (1997/98, 2000/01, and 2007/08; the Czech post-MONICA Study) keeping, by law, a list of all the insured individuals. Health insurance is mandatory for all Czech citizens and is paid for by the employer/employee or by the government for children, the retired and unemployed persons.

In the last survey, conducted in 2016/17, the random selection was performed from registers of five major health insurance companies operating in the Czech Republic and covering 85% of the total population.

The Czech MONICA and Czech post-MONICA studies were approved by the Ethics Committee of the Institute for Clinical and Experimental Medicine and Thomayer Hospital, Prague, Czech Republic. All participants provided their informed consent.

## Screening examination

The methods used were described in detail elsewhere [10]. In brief, the examination consisted of a physician-completed questionnaire. Currently prescribed drugs were recorded and verified (if possible) against drug containers.

Height and body weight were measured, BP measurement was performed consistently on the right arm (supported at the heart level), in the sitting position, after at least a 5-minute rest, using standard mercury sphygmomanometers and correctly sized cuffs. Blood pressure values were recorded to the nearest 2 mmHg. In 1985, 1988, and 1992, two consecutive BP measurements were performed with their mean values used for the longitudinal trend analysis. In 1997/98, 2000/01, 2007/08 and 2016/17 three consecutive BP measurements were obtained; however, for longitudinal trend analysis, only the mean of the first two readings was used.

A gentle venous blood sampling was performed in the sitting position after at least a 12-hour fast. The obtained samples were centrifuged at 1,500 G and frozen thereafter.

## Laboratory analysis

Lipid parameters in all the seven surveys were analyzed in the same Lipid Laboratory of the Institute for Clinical and Experimental Medicine, serving as the WHO Reference Laboratory throughout the WHO MONICA project. In 1985–1992, total cholesterol was determined using an enzymatic method and CHOD-PAP kits (Boehringer, Mannheim, Germany). HDL-cholesterol was also assessed by enzymatic methods after precipitation of serum apolipoprotein B-containing lipoproteins with sodium phosphotungstate. Since 1997, lipid parameters have been determined using a fully automated enzymatic method (COBAS MIRA S analyzer, Roche Diagnostics, Indianapolis, Indiana, USA) with enzymatic kits produced by the same manufacturer.

Accuracy of analysis has been continuously monitored and tested by the Centers for Disease Control and Prevention (Atlanta, Georgia, USA); all analyses of total cholesterol and HDL-cholesterol were within the limit ± 2%.

## Definition of major risk factors

Smoking was assessed using the WHO definition, with a current smoker defined as smoking at least one cigarette per day. Obesity was defined as BMI $\geq 30$ kg/m$^2$ for both sexes.

Hypertension was defined as a mean SBP $\geq 140$ mmHg, and/or a mean DBP $\geq 90$ mmHg, or current treatment with antihypertensive drugs. Awareness of hypertension was defined as an individual having reported a previous diagnosis of hypertension or current use of

antihypertensive medication. Treatment of hypertension was defined as current use of prescribed medication affecting BP. Hypertension control was defined as an individual receiving drug treatment for hypertension and achieving SBP <140 mmHg and DBP <90 mmHg.

Dyslipidemia was defined as total cholesterol ≥5 mmol/L (~190 mg/dL) or HDL-cholesterol <1 mmol/L (~40 mg/dL) in men and <1.2 mmol/L (~45 mg/dL) in women or use of lipid-lowering drugs [12].

### Statistical analysis

Statistical analyses were performed using JMP® 11.0.0 statistical software (2013, SAS Institute Inc.). Trends for means were tested by linear contrast in one-way ANOVA and trends for percentage by Cochran Armitage trend test of proportions. If necessary, Bonferroni correction for the adjustment of $p$ values was applied. Three-way ANOVA and logistic regression were used to determine a possible influence of sex, age groups and the year(s) of examination on risk factors.

## Results

### Population sample characteristics and response rates

The total number in all seven independent cross-sectional surveys included 15,656 individuals of European descent, with consistently slightly higher response rates in women (Table 1). There was a significant downward linear trend in the response rates in both sexes with a sharp decline between the last two surveys, particularly in the youngest age groups.

### Trends in cigarette smoking

Cigarette smoking declined significantly in males, by 21.1% (from 45.0 to 23.9%, $p<0.001$), between 1985 and 2016/17. There was a significant downward trend observed in all male age groups except the oldest one (aged 55–64 years with no significant change).

**Table 1. Survey sample sizes and response rates.**

|  | 1985 | 1988 | 1992 | 1997/98 | 2000/01 | 2007/08 | 2016/17 | *p* for trend |
|---|---|---|---|---|---|---|---|---|
| Total | 2,570 | 2,768 | 2,343 | 1,990 | 2,055 | 2,246 | 1,684 | |
| Mean age, yrs | 44.9 ± 11.38 | 45.1 ± 11.26 | 44.7 ± 10.87 | 45.6 ± 10.64 | 46.2 ± 11.9 | 47.1 ± 11.46 | 47.8 ± 10.85 | <0.001 |
| Men | 1,253 | 1,357 | 1,134 | 969 | 1,003 | 1,102 | 788 | |
| Mean age, yrs | 45.0 ± 11.39 | 45.3 ± 11.29 | 44.6 ± 10.76 | 45.8 ± 10.63 | 46.7 ± 11.07 | 47.9 ± 11.65 | 48.0 ± 10.83 | <0.001 |
| Response rate (%) | 81.5 | 85.5 | 73.2 | 63.2 | 62.0 | 62.1 | 43.1 | <0.001 |
| Age group, n (%) | | | | | | | | |
| 25–34 yrs | 307 (24.5) | 322 (23.7) | 246 (21.7) | 194 (20.0) | 187 (18.6) | 208 (18.9) | 116 (14.7) | <0.001 |
| 35–44 yrs | 296 (23.6) | 323 (23.8) | 350 (30.9) | 230 (23.7) | 230 (22.9) | 251 (22.8) | 198 (25.1) | ns |
| 45–54 yrs | 334 (26.7) | 361 (26.6) | 310 (27.3) | 332 (34.3) | 295 (29.4) | 231 (21.0) | 210 (26.7) | ns |
| 55–64 yrs | 316 (25.2) | 351 (25.9) | 228 (20.1) | 213 (22.0) | 291 (29.0) | 412 (37.4) | 264 (33.5) | ns |
| Women | 1,317 | 1,411 | 1,209 | 1,021 | 1,052 | 1,144 | 896 | |
| Mean age, yrs | 44.9 ± 11.38 | 44.9 ± 11.24 | 44.9 ± 10.97 | 45.3 ± 10.65 | 45.8 ± 11.10 | 46.4 ± 11.23 | 47.6 ± 10.88 | <0.001 |
| Response rate (%) | 85.0 | 88.4 | 76.7 | 66.4 | 63.8 | 63.1 | 48.6 | <0.001 |
| Age group, n (%) | | | | | | | | |
| 25–34 yrs | 322 (24.4) | 342 (24.2) | 266 (22.0) | 212 (20.8) | 213 (20.2) | 235 (20.5) | 147 (16.4) | <0.001 |
| 35–44 yrs | 340 (25.8) | 369 (26.2) | 356 (29.4) | 266 (26.1) | 276 (26.2) | 284 (24.8) | 204 (22.8) | ns |
| 45–54 yrs | 343 (26.0) | 360 (25.5) | 311 (25.7) | 326 (31.9) | 285 (27.1) | 299 (26.1) | 282 (31.5) | ns |
| 55–64 yrs | 312 (23.7) | 340 (24.1) | 276 (22.8) | 217 (21.3) | 278 (26.4) | 326 (28.5) | 263 (29.4) | ns |

There was no change in the overall proportion of female smokers over the analyzed period, varying from 20.9 to 25.9%. A significant decrease in cigarette smokers was observed in the two younger female age groups (25–44 years), while the group 45–54 years did not change their smoking habits and the oldest female group (55–64 years) showed a significant increase in the prevalence of smoking (from 9.3 to 20.3%; $p < 0.001$) (Table 2).

Employing logistic regression, the interaction between the year of examination and age groups as well as between sex and the year of examination was found to be significant ($p < 0.001$; Fig 1A).

Daily consumption of cigarettes decreased significantly only in males (from 17.8 ± 7.8 to 14.8 ± 8.3; $p < 0.05$).

## Trends in anthropometric parameters

A significant upward trend in height and body weight was observed in both genders, whereas BMI increased only in males over the period of 30 years (Table 3). The increase in body weight in males was alarming (from 81.7 ± 12.8 to 92.1 ± 16.8 kg; $p < 0.001$) and consistent across all the age groups. Contrarily, in women, there was no change in the entire group and even a significant decline of BMI in the age group 45–54 years (from 28.6 ± 4.9 to 27.4 ± 5.6 kg; $p = 0.007$). Using three-way ANOVA, the interaction between sex and age groups as well as between sex and the year of examination was found to be significant ($p < 0.001$; Fig 1B).

## Trends in blood pressure and in prevalence, awareness, treatment, and control of hypertension

A downward trend in SBP and DBP was observed in both genders with a greater decline in females (males: from 135.8 ± 19.2/85.9 ± 11.0 to 131.1 ± 14.9/84.7 ± 9.1 mmHg; $p < 0.001$; females: from 131.6 ± 20.9/82.5 ± 11.3 to 124.8 ± 16.9/80.0 ± 9.4 mmHg; $p < 0.001$) (Table 4). In males, significant changes both in SBP and DBP were found only in individuals aged over

**Table 2. Smoking habits between 1985 and 2016/17 in six districts of the Czech Republic.**

| | 1985 | 1988 | 1992 | 1997/98 | 2000/01 | 2007/08 | 2016/17 | *p* for trend |
|---|---|---|---|---|---|---|---|---|
| Males | | | | | | | | |
| Smokers, % | | | | | | | | |
| Total, 25–64 yrs | 564 (45.0) | 572 (42.2) | 450 (39.7) | 356 (36.8) | 355 (35.5) | 336 (30.5) | 187 (23.9) | <0.001 |
| 25–34 yrs | 164 (53.4) | 165 (51.2) | 104 (42.3) | 84 (43.3) | 74 (39.6) | 71 (34.1) | 34 (29.3) | <0.001 |
| 35–44 yrs | 163 (55.1) | 156 (48.3) | 157 (44.9) | 84 (36.5) | 87 (38.0) | 83 (33.1) | 52 (26.4) | <0.001 |
| 45–54 yrs | 132 (39.5) | 156 (43.2) | 130 (41.9) | 136 (41.0) | 122 (41.4) | 70 (30.3) | 43 (20.7) | <0.001 |
| 55–64 yrs | 105 (33.2) | 95 (27.1) | 59 (25.9) | 52 (24.5) | 72 (24.8) | 112 (27.2) | 58 (22.1) | ns |
| Nr. of cig., % | 17.8 ± 7.8 | 15.4 ± 7.4 | 15.8 ± 7.5 | 16.6 ± 9.0 | 17.2 ± 8.5 | 15.2 ± 8.9 | 14.8 ± 8.3 | <0.030 |
| Females | | | | | | | | |
| Smokers, % | | | | | | | | |
| Total, 25–64 yrs | 315 (23.9) | 339 (24.0) | 280 (23.2) | 264 (25.9) | 244 (23.2) | 267 (23.3) | 187 (20.9) | ns |
| 25–34 yrs | 115 (35.7) | 106 (31.0) | 69 (26.0) | 60 (28.4) | 45 (21.1) | 52 (22.0) | 38 (25.9) | <0.001 |
| 35–44 yrs | 116 (34.1) | 126 (34.2) | 108 (30.3) | 86 (32.3) | 79 (28.6) | 62 (21.9) | 36 (17.7) | <0.001 |
| 45–54 yrs | 55 (16.0) | 72 (20.0) | 79 (25.4) | 82 (25.2) | 72 (25.3) | 80 (26.8) | 60 (21.3) | ns |
| 55–64 yrs | 29 (9.3) | 35 (10.3) | 24 (8.7) | 36 (16.7) | 48 (17.3) | 73 (22.4) | 53 (20.3) | <0.001 |
| Nr. of cig., % | 11.0 ± 6.3 | 10.2 ± 5.4 | 10.4 ± 5.7 | 11.1 ± 6.1 | 11.4 ± 6.5 | 9.9 ± 5.9 | 8.9 ± 5.3 | ns |

Please note the numbers for the survey population may differ slightly from those given in Table 1 as the information about smoking habits was not available for all individuals.

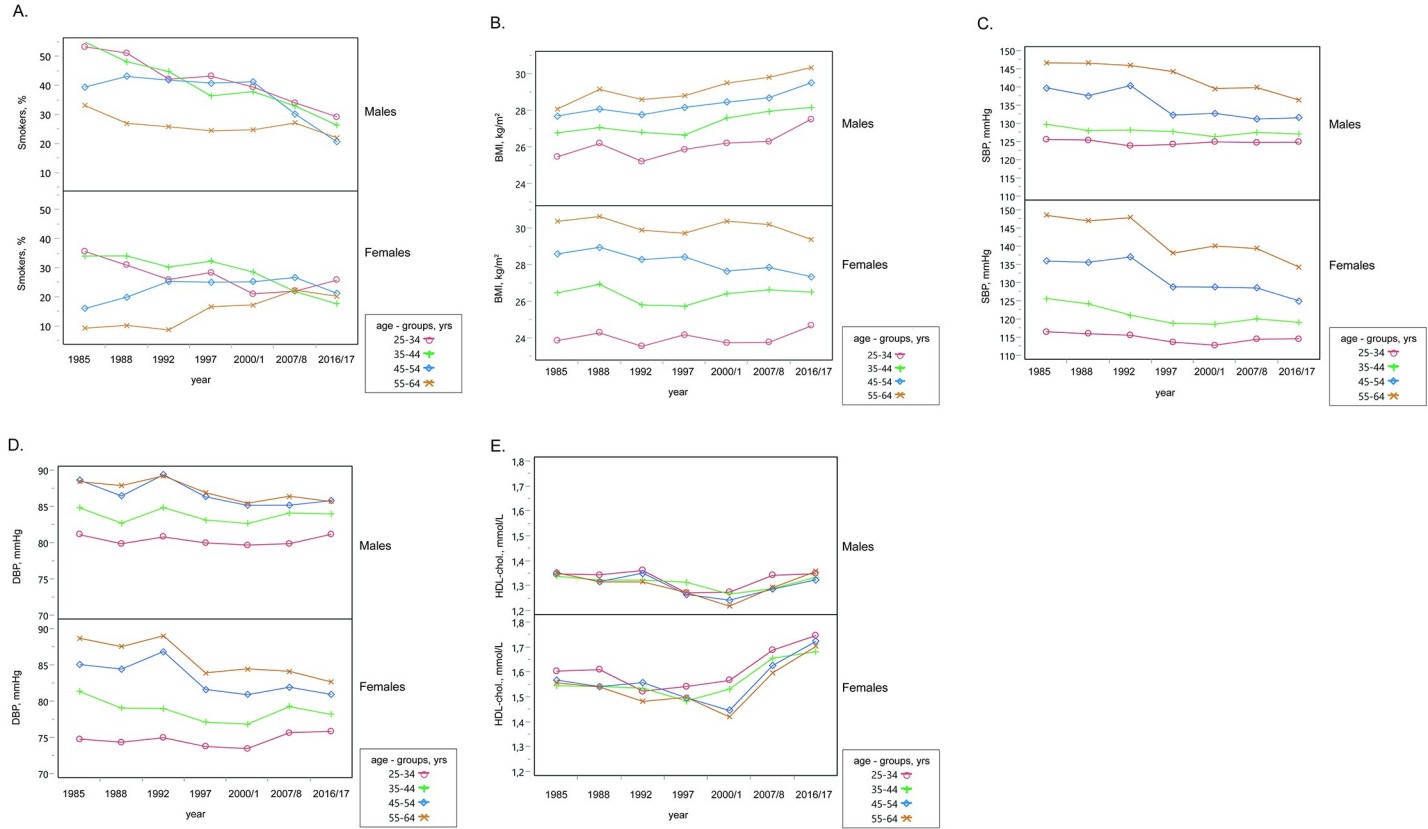

**Fig 1.** A. Smoking habits by age groups and gender between 1985 and 2016/17 in 6 districts of the Czech Republic. B. Body mass index (BMI) by age groups and gender between 1985 and 2016/17 in 6 districts of the Czech Republic. C. Systolic blood pressure (SBP) by age groups and gender between 1985 and 2016/17 in 6 districts of the Czech Republic. D. Diastolic blood pressure (DBP) by age groups and gender between 1985 and 2016/17 in 6 districts of the Czech Republic. E. HDL-cholesterol by age groups and gender between 1985 and 2016/17 in 6 districts of the Czech Republic.

45 years, whereas in females, all the changes were significant except for DBP in the youngest age group. Using three-way ANOVA, the interaction between sex and age groups as well as between sex and the year of examination was found to be significant (Fig 1C and 1D).

Over the period of 30 years, the prevalence of hypertension declined in the entire study population (from 47.1% in 1985 to 41.5% in 2016/17; $p$ <0.001) due to the declining prevalence in females (from 42.5% in 1985 to 33.5% in 2016/17; $p$<0.001; the decline was significant in all age groups, except for the youngest group). There was no change in the prevalence of hypertension in the overall male population, achieving 50.6% during the last survey. There was, however, a significant decline in the prevalence of hypertension in the two male middle-aged groups (35–44 and 45–54 years) (Table 4).

Awareness of hypertension increased in both genders, remaining higher in females over the entire study period (males: from 41.4% in 1985 to 74.6% in 2016/17; $p$<0.001; females: from 58.9% in 1985 to 77.7% in 2016/17; $p$<0.001). Furthermore, the increase in awareness of hypertension was found in both genders in all age groups except for the youngest groups.

The number of individuals treated by antihypertensive drugs increased significantly in both genders, again showing consistently higher rates in females (males: from 21.1 to 60.9%; $p$<0.001; females: from 38.9 to 64.8%; $p$<0.001) (Table 4). The proportion of individuals treated by antihypertensive drugs increased significantly in all age groups, with the exception of the youngest groups, in both genders.

**Table 3. Anthropometric parameters between 1985 and 2016/17 in six districts of the Czech Republic.**

|  | 1985 | 1988 | 1992 | 1997/98 | 2000/01 | 2007/08 | 2016/17 | p for trend |
|---|---|---|---|---|---|---|---|---|
| Males |  |  |  |  |  |  |  |  |
| Height, cm | 173.8 ± 6.8 | 174.3 ± 7.0 | 174.8 ± 7.1 | 175.6 ± 7.1 | 175.3 ± 7.0 | 177.6 ± 7.0 | 177.7 ± 7.3 | <0.001 |
| Body weight, kg | 81.7 ± 12.8 | 84.2 ± 12.8 | 82.8 ± 12.8 | 84.8 ± 13.1 | 86.5 ± 14.6 | 90.0 ± 15.8 | 92.1 ± 16.8 | <0.001 |
| BMI, kg/m$^2$ |  |  |  |  |  |  |  |  |
| Total, 25–64 yrs | 27.0 ± 4.0 | 27.7 ± 3.8 | 27.1 ± 3.8 | 27.5 ± 3.8 | 28.1 ± 4.4 | 28.5 ± 4.6 | 29.2 ± 5.1 | <0.001 |
| 25–34 yrs | 25.5 ± 3.4 | 26.2 ± 3.3 | 25.2 ± 3.2 | 25.9 ± 3.2 | 26.2 ± 4.3 | 26.3 ± 4.3 | 27.5 ± 4.9 | 0.011 |
| 35–44 yrs | 26.8 ± 3.8 | 27.1 ± 3.7 | 26.8 ± 3.6 | 26.7 ± 3.3 | 27.6 ± 3.9 | 28.0 ± 4.5 | 28.2 ± 4.8 | <0.001 |
| 45–54 yrs | 27.7 ± 3.8 | 28.1 ± 3.7 | 27.8 ± 3.6 | 28.2 ± 3.9 | 28.5 ± 4.2 | 28.7 ± 4.4 | 29.5 ± 5.0 | <0.001 |
| 55–64 yrs | 28.1 ± 4.3 | 29.2 ± 3.9 | 28.6 ± 4.1 | 28.8 ± 3.8 | 29.5 ± 4.6 | 29.8 ± 4.6 | 30.4 ± 5.3 | <0.001 |
| BMI ≥ 30, kg/m$^2$ (%) | 246 (19.7) | 343 (25.3) | 225 (19.9) | 244 (25.2) | 295 (29.5) | 370 (33.6) | 297 (37.7) | <0.001 |
| Females |  |  |  |  |  |  |  |  |
| Height, cm | 161.2 ± 6.3 | 161.5 ± 6.4 | 162.2 ± 6.4 | 162.5 ± 6.4 | 162.7 ± 6.3 | 164.6 ± 6.4 | 164.4 ± 6.0 | <0.001 |
| Body weight, kg | 70.8 ± 13.6 | 72.1 ± 13.8 | 70.7 ± 13.9 | 71.5 ± 14.2 | 72.1 ± 15.0 | 74.1 ± 16.2 | 73.8 ± 16.3 | <0.001 |
| BMI, kg/m$^2$ |  |  |  |  |  |  |  |  |
| Total, 25–64 yrs | 27.3 ± 5.4 | 27.7 ± 5.4 | 26.9 ± 5.3 | 27.1 ± 5.5 | 27.3 ± 5.7 | 27.3 ± 5.7 | 27.3 ± 6.0 | ns |
| 25–34 yrs | 23.9 ± 4.1 | 24.3 ± 3.9 | 23.6 ± 4.0 | 24.2 ± 4.6 | 23.8 ± 4.1 | 23.8 ± 4.8 | 24.7 ± 5.4 | ns |
| 35–44 yrs | 26.5 ± 4.7 | 26.9 ± 4.9 | 25.8 ± 4.9 | 25.8 ± 4.9 | 26.4 ± 5.5 | 26.6 ± 5.7 | 26.5 ± 6.0 | ns |
| 45–54 yrs | 28.6 ± 4.9 | 29.0 ± 5.0 | 28.3 ± 5.5 | 28.4 ± 5.6 | 27.7 ± 5.1 | 27.9 ± 5.7 | 27.4 ± 5.6 | 0.007 |
| 55–64 yrs | 30.4 ± 5.4 | 30.7 ± 5.4 | 29.9 ± 5.1 | 29.7 ± 5.0 | 30.4 ± 5.9 | 30.2 ± 5.9 | 29.4 ± 6.2 | ns |
| BMI ≥ 30, kg/m$^2$ (%) | 367 (28.0) | 423 (30.0) | 308 (25.5) | 270 (26.5) | 292 (27.8) | 344 (28.1) | 247 (27.6) | ns |

BMI, body mass index

Overall hypertension control increased significantly over the period of 30 years (from 3.9 to 32.9%; p <0.001), being consistently more effective in females. The improvement in hypertension control was consistent across all the age groups in both genders except for the youngest one in females.

## Trends in lipid parameters

Over the period of 31/32 years, a significant downward trend in total cholesterol was found in both genders (males: from 6.21 ± 1.29 to 5.30 ± 1.05; p <0.001; females: from 6.18 ± 1.26 to 5.31 ± 1.00 mmol/L; p <0.001) (Table 5). Over the same period, there was also a small but significant decline in HDL-cholesterol in males (from 1.35 ± 0.36 to 1.34 ± 0.36 mmol/L; p <0.001) and no significant overall change in females (from 1.57 ± 0.36 to 1.71 ± 0.43 mmol/L; ns). Nevertheless, a significant increase in HDL-cholesterol was observed in the two youngest female age groups. All the main effects (sex, age groups, year of examination) were found significant (Fig 1E; three-way ANOVA). There was also a significant decrease in non-HDL-cholesterol (males: from 4.86 ± 1.35 to 3.96 ± 1.09 mmol/L; p <0.001; females: 4.61 ± 1.29 to 3.60 ± 1.03 mmol/L; p <0.001) and in the total to HDL-cholesterol ratio (males: from 4.94 ± 1.83 to 4.25 ± 1.55; p <0.001; females: from 4.14 ± 1.32 to 3.30 ± 1.12; p <0.001) in both genders (Table 5).

The information on lipid-lowering drugs was only available since the fourth survey (1997/98), indicating a significant increase in their use in both genders (males: from 4.8% in 1997/1998 to 14.6% in 2016/17; p <0.0001; females: from 4.3% in 1997/1998 to 10.0% in 2016/17; p <0.0001). Fibrates were the most frequently used lipid-lowering drugs (accounting for 81.1% of all the lipid-lowering drugs) in 1997/98, while statins have been the most prescribed ones nowadays (78.8%).

**Table 4. Blood pressure (mean ± SD), prevalence, awareness, treatment, and control of hypertension between 1985 and 2016/17 in six districts of the Czech Republic.**

| | 1985 | 1988 | 1992 | 1997/98 | 2000/01 | 2007/08 | 2016/17 | p for trend |
|---|---|---|---|---|---|---|---|---|
| Males | | | | | | | | |
| SBP, mmHg | | | | | | | | |
| Total, 25–64 yrs | 135.8 ± 19.2 | 134.9 ± 19.2 | 134.2 ± 20.0 | 132.3 ± 16.9 | 131.9 ± 16.8 | 132.5 ± 17.3 | 131.1 ± 14.9 | <0.001 |
| 25–34 yrs | 125.7 ± 14.6 | 125.5 ± 13.6 | 123.9 ± 13.4 | 124.3 ± 11.4 | 125.0 ± 15.3 | 124.8 ± 11.7 | 124.9 ± 12.6 | ns |
| 35–44 yrs | 129.9 ± 15.7 | 128.1 ± 15.5 | 128.2 ± 15.4 | 127.9 ± 13.7 | 126.4 ± 13.9 | 127.6 ± 13.9 | 127.1 ± 13.1 | ns |
| 45–54 yrs | 139.9 ± 18.9 | 137.7 ± 19.1 | 140.5 ± 20.1 | 132.4 ± 15.9 | 132.8 ± 16.0 | 131.3 ± 16.6 | 131.6 ± 14.2 | <0.001 |
| 55–64 yrs | 146.7 ± 19.4 | 146.7 ± 19.9 | 146.0 ± 22.8 | 144.3 ± 19.1 | 139.7 ± 17.1 | 139.9 ± 18.7 | 136.5 ± 15.6 | <0.001 |
| DBP, mmHg | | | | | | | | |
| Total, 25–64 yrs | 85.9 ± 11.0 | 84.4 ± 11.0 | 86.1 ± 11.4 | 84.5 ± 10.0 | 83.7 ± 9.7 | 84.4 ± 10.1 | 84.7 ± 9.1 | <0.001 |
| 25–34 yrs | 81.1 ± 10.0 | 79.9 ± 10.3 | 80.8 ± 9.7 | 80.0 ± 8.7 | 79.7 ± 9.7 | 79.9 ± 8.5 | 81.2 ± 9.6 | ns |
| 35–44 yrs | 84.8 ± 9.8 | 82.7 ± 10.3 | 84.9 ± 10.1 | 83.2 ± 9.5 | 82.7 ± 9.2 | 84.1 ± 10.4 | 84.0 ± 8.5 | ns |
| 45–54 yrs | 88.7 ± 11.2 | 86.5 ± 11.1 | 86.4 ± 11.4 | 86.4 ± 9.6 | 85.2 ± 9.2 | 85.2 ± 8.9 | 85.8 ± 9.2 | <0.001 |
| 55–64 yrs | 88.5 ± 11.2 | 87.9 ± 10.4 | 89.3 ± 11.7 | 87.0 ± 10.8 | 85.5 ± 9.6 | 86.5 ± 10.6 | 85.7 ± 8.9 | <0.001 |
| Prevalence of HT, % | | | | | | | | |
| Total, 25–64 yrs | 650 (51.9) | 639 (47.1) | 508 (44.8) | 408 (42.1) | 457 (45.6) | 553 (50.2) | 399 (50.6) | ns |
| 25–34 yrs | 85 (27.7) | 67 (20.8) | 46 (18.7) | 35 (18.0) | 34 (18.2) | 43 (20.7) | 26 (22.4) | ns |
| 35–44 yrs | 123 (41.6) | 119 (36.8) | 126 (36.0) | 61 (26.5) | 69 (30.0) | 82 (32.7) | 66 (33.3) | 0.049 |
| 45–54 yrs | 208 (62.3) | 198 (54.9) | 186 (60.0) | 161 (48.5) | 147 (49.8) | 122 (52.8) | 107 (51.0) | 0.009 |
| 55–64 yrs | 234 (74.1) | 255 (72.7) | 150 (65.8) | 151 (70.9) | 207 (71.1) | 306 (74.3) | 200 (75.8) | ns |
| Awareness of HT, % | 269 (41.4) | 320 (50.1) | 232 (45.7) | 230 (56.4) | 284 (62.1) | 378 (68.4) | 290 (74.6) | <0.001 |
| Medication for HT,% | 137 (21.1) | 197 (30.8) | 123 (24.2) | 151 (37.0) | 191 (41.8) | 322 (58.2) | 241 (60.9) | <0.001 |
| Control of HT, % | 18 (2.8) | 33 (5.2) | 14 (2.8) | 50 (12.3) | 60 (13.1) | 135 (24.4) | 119 (29.8) | <0.001 |
| Females | | | | | | | | |
| SBP, mmHg | | | | | | | | |
| Total, 25–64 yrs | 131.6 ± 20.9 | 130.7 ± 20.9 | 130.2 ± 22.0 | 125.2 ± 18.1 | 125.9 ± 18.8 | 126.7 ± 19.2 | 124.8 ± 16.9 | <0.001 |
| 25–34 yrs | 116.6 ± 13.7 | 116.0 ± 12.2 | 115.6 ± 13.3 | 113.7 ± 9.9 | 112.9 ± 11.0 | 114.5 ± 12.2 | 114.7 ± 12.4 | 0.004 |
| 35–44 yrs | 125.8 ± 15.8 | 124.3 ± 16.0 | 121.1 ± 16.0 | 118.9 ± 14.6 | 118.7 ± 12.7 | 120.1 ± 15.5 | 119.2 ± 13.4 | <0.001 |
| 45–54 yrs | 136.0 ± 19.1 | 135.7 ± 18.4 | 137.1 ± 21.0 | 129.0 ± 17.9 | 128.9 ± 17.8 | 128.6 ± 17.0 | 125.1 ± 16.3 | <0.001 |
| 55–64 yrs | 148.6 ± 19.9 | 147.1 ± 21.7 | 148.0 ± 21.2 | 138.2 ± 18.3 | 140.2 ± 19.1 | 139.5 ± 20.2 | 134.4 ± 17.0 | <0.001 |
| DBP, mmHg | | | | | | | | |
| Total, 25–64 yrs | 82.5 ± 11.3 | 81.4 ± 11.2 | 82.5 ± 12.1 | 79.3 ± 9.8 | 79.3 ± 9.8 | 80.6 ± 9.6 | 80.0 ± 9.4 | <0.001 |
| 25–34 yrs | 74.8 ± 9.2 | 74.4 ± 8.7 | 75.0 ± 9.1 | 73.8 ± 7.7 | 73.5 ± 7.9 | 75.7 ± 8.5 | 75.9 ± 9.3 | ns |
| 35–44 yrs | 81.4 ± 9.9 | 79.1 ± 10.1 | 79.0 ± 10.5 | 77.1 ± 9.0 | 76.9 ± 8.8 | 79.3 ± 8.8 | 78.2 ± 9.0 | <0.001 |
| 45–54 yrs | 85.1 ± 10.7 | 84.5 ± 10.5 | 86.9 ± 11.9 | 81.7 ± 10.0 | 80.9 ± 8.8 | 82.0 ± 9.3 | 81.0 ± 9.2 | <0.001 |
| 55–64 yrs | 88.7 ± 10.3 | 87.6 ± 10.6 | 89.1 ± 11.5 | 83.9 ± 8.8 | 84.5 ± 10.0 | 84.2 ± 9.3 | 82.6 ± 8.8 | <0.001 |
| Prevalence of HT, % | | | | | | | | |
| Total, 25–64 yrs | 560 (42.5) | 552 (39.1) | 460 (38.0) | 323 (31.6) | 347 (33.0) | 426 (37.3) | 300 (33.5) | <0.001 |
| 25–34 yrs | 30 (9.3) | 27 (7.9) | 22 (8.3) | 7 (3.3) | 10 (4.7) | 16 (6.8) | 7 (4.8) | ns |
| 35–44 yrs | 98 (28.8) | 88 (23.9) | 67 (18.8) | 41 (15.4) | 37 (13.4) | 62 (21.9) | 26 (12.8) | <0.001 |
| 45–54 yrs | 186 (54.2) | 180 (50.0) | 166 (53.4) | 134 (41.1) | 110 (38.6) | 124 (41.5) | 111 (39.4) | <0.001 |
| 55–64 yrs | 246 (78.9) | 257 (75.6) | 205 (74.3) | 141 (65.0) | 190 (68.4) | 224 (68.7) | 156 (59.3) | <0.001 |
| Awareness of HT, % | 330 (58.9) | 330 (59.8) | 255 (55.4) | 221 (68.4) | 256 (73.8) | 304 (71.4) | 233 (77.7) | <0.001 |
| Medication for HT,% | 218 (38.9) | 233 (42.2) | 159 (34.6) | 187 (57.9) | 205 (59.1) | 251 (58.9) | 193 (64.8) | <0.001 |
| Control of HT, % | 29 (5.2) | 51 (9.2) | 28 (6.1) | 70 (21.7) | 77 (22.2) | 106 (24.9) | 111 (37.0) | <0.001 |

p = statistical significance for linear trend

SBP, systolic blood pressure, DBP, diastolic blood pressure, HT, hypertension

**Table 5. Lipid parameters between 1985 and 2016/17 in six districts of the Czech Republic.**

| | 1985 | 1988 | 1992 | 1997/98 | 2000/01 | 2007/08 | 2016/17 | P for trend |
|---|---|---|---|---|---|---|---|---|
| Males | | | | | | | | |
| TC, mmol/L | | | | | | | | |
| Total, 25–64 yrs | 6.21 ± 1.29 | 6.29 ± 1.21 | 5.98 ± 1.30 | 5.65 ± 1.15 | 5.88 ± 1.08 | 5.29 ± 1.10 | 5.30 ± 1.05 | <0.001 |
| 25–34 yrs | 5.71 ± 1.33 | 5.87 ± 1.14 | 5.31 ± 1.03 | 5.15 ± 1.10 | 5.26 ± 1.00 | 4.88 ± 0.90 | 4.84 ± 0.81 | <0.001 |
| 35–44 yrs | 6.34 ± 1.34 | 6.20 ± 1.11 | 6.03 ± 1.17 | 5.70 ± 1.11 | 5.85 ± 1.08 | 5.38 ± 1.03 | 5.39 ± 0.98 | <0.001 |
| 45–54 yrs | 6.39 ± 1.18 | 6.60 ± 1.30 | 6.34 ± 1.38 | 5.77 ± 1.10 | 6.13 ± 1.06 | 5.43 ± 1.15 | 5.50 ± 1.22 | <0.001 |
| 55–64 yrs | 6.37 ± 1.17 | 6.42 ± 1.15 | 6.15 ± 1.37 | 5.86 ± 1.15 | 6.06 ± 1.01 | 5.37 ± 1.16 | 5.29 ± 1.00 | <0.001 |
| HDL-C, mmol/L | | | | | | | | |
| Total, 25–64 yrs | 1.35 ± 0.36 | 1.33 ± 0.32 | 1.34 ± 0.49 | 1.28 ± 0.32 | 1.25 ± 0.33 | 1.30 ± 0.34 | 1.34 ± 0.36 | <0.001 |
| 25–34 yrs | 1.35 ± 0.31 | 1.34 ± 0.31 | 1.36 ± 0.37 | 1.27 ± 0.29 | 1.27 ± 0.35 | 1.34 ± 0.34 | 1.35 ± 0.34 | ns |
| 35–44 yrs | 1.34 ± 0.35 | 1.32 ± 0.32 | 1.32 ± 0.37 | 1.32 ± 0.32 | 1.27 ± 0.35 | 1.29 ± 0.33 | 1.34 ± 0.35 | ns |
| 45–54 yrs | 1.35 ± 0.38 | 1.32 ± 0.30 | 1.35 ± 0.62 | 1.27 ± 0.32 | 1.24 ± 0.33 | 1.29 ± 0.32 | 1.33 ± 0.38 | 0.014 |
| 55–64 yrs | 1.35 ± 0.38 | 1.32 ± 0.33 | 1.32 ± 0.56 | 1.28 ± 0.33 | 1.22 ± 0.31 | 1.29 ± 0.36 | 1.36 ± 0.37 | 0.006 |
| Non HDL-C, mmol/L | 4.86 ± 1.35 | 4.96 ± 1.26 | 4.65 ± 1.33 | 4.36 ± 1.16 | 4.63 ± 1.11 | 3.97 ± 1.10 | 3.96 ± 1.09 | <0.001 |
| TC/HDL-C | 4.94 ± 1.83 | 5.01 ± 1.65 | 4.83 ± 1.66 | 4.66 ± 1.46 | 5.01 ± 1.53 | 4.32 ± 1.39 | 4.25 ± 1.55 | <0.001 |
| Dyslipidemia*, % (N/T)* | 87.5 (1,093/1,249) | 89.3 (1,208/1,352) | 83.5 (945/1,132) | 77.6 (751/968) | 85.0 (853/1,003) | 73.8 (804/1,089) | 74.8 (589/787) | <0.001 |
| Females | | | | | | | | |
| TC, mmol/L | | | | | | | | |
| Total, 25–64 yrs | 6.18 ± 1.26 | 6.22 ± 1.21 | 5.95 ± 1.29 | 5.53 ± 1.21 | 5.82 ± 1.13 | 5.30 ± 1.06 | 5.31 ± 1.00 | <0.001 |
| 25–34 yrs | 5.46 ± 1.13 | 5.58 ± 1.01 | 5.15 ± 0.92 | 4.76 ± 0.83 | 5.01 ± 0.88 | 4.68 ± 0.86 | 4.73 ± 0.86 | <0.001 |
| 35–44 yrs | 5.93 ± 1.06 | 5.94 ± 1.05 | 5.55 ± 1.08 | 5.23 ± 1.04 | 5.53 ± 0.93 | 5.02 ± 0.91 | 4.94 ± 0.82 | <0.001 |
| 45–54 yrs | 6.53 ± 1.18 | 6.48 ± 1.15 | 6.26 ± 1.17 | 5.73 ± 1.13 | 6.00 ± 1.00 | 5.55 ± 1.05 | 5.48 ± 0.95 | <0.001 |
| 55–64 yrs | 6.78 ± 1.22 | 6.89 ± 1.22 | 6.87 ± 1.26 | 6.36 ± 1.23 | 6.51 ± 1.14 | 5.77 ± 1.05 | 5.73 ± 1.00 | <0.001 |
| HDL-C, mmol/L | | | | | | | | |
| Total, 25–64 yrs | 1.57 ± 0.36 | 1.56 ± 0.34 | 1.53 ± 0.46 | 1.50 ± 0.36 | 1.49 ± 0.38 | 1.64 ± 0.38 | 1.71 ± 0.43 | ns |
| 25–34 yrs | 1.60 ± 0.35 | 1.61 ± 0.34 | 1.52 ± 0.39 | 1.54 ± 0.35 | 1.57 ± 0.38 | 1.69 ± 0.35 | 1.75 ± 0.39 | <0.05 |
| 35–44 yrs | 1.55 ± 0.34 | 1.54 ± 0.33 | 1.54 ± 0.38 | 1.49 ± 0.36 | 1.53 ± 0.39 | 1.66 ± 0.40 | 1.68 ± 0.41 | 0.006 |
| 45–54 yrs | 1.57 ± 0.36 | 1.54 ± 0.32 | 1.56 ± 0.64 | 1.50 ± 0.36 | 1.45 ± 0.37 | 1.63 ± 0.42 | 1.72 ± 0.45 | ns |
| 55–64 yrs | 1.56 ± 0.39 | 1.54 ± 0.36 | 1.48 ± 0.35 | 1.50 ± 0.38 | 1.43 ± 0.38 | 1.60 ± 0.36 | 1.70 ± 0.46 | ns |
| Non HDL-C, mmol/L | 4.61 ± 1.29 | 4.66 ± 1.25 | 4.44 ± 1.32 | 4.03 ± 1.24 | 4.33 ± 1.18 | 3.65 ± 1.12 | 3.60 ± 1.03 | <0.001 |
| TC/HDL-C | 4.14 ± 1.32 | 4.18 ± 1.27 | 4.16 ± 1.39 | 3.89 ± 1.30 | 4.17 ± 1.38 | 3.42 ± 1.15 | 3.30 ± 1.12 | <0.001 |
| Dyslipidemia*, % (N/T) | 87.7 (1,152/1,314) | 88.0 (1,239/1,408) | 80.6 (974/1,209) | 70.4 (718/1,020) | 80.1 (843/1,052) | 66.0 (737/1,117) | 69.9 (626/895) | <0.001 |

TC, total cholesterol; HDL-C, HDL-cholesterol; N/T, number of individuals with dyslipidemia over the survey population

Dyslipidemia was defined as total cholesterol ≥5 mmol/L (~190 mg/dL) or HDL-cholesterol <1 mmol/L (~40 mg/dL) in men and <1.2 mmol/L (~45 mg/dL) in women or use of lipid-lowering drugs.

Please note the numbers for the survey population may differ slightly from those given in Table 1 as lipid analysis was not available for all individuals.

Overall, the prevalence of dyslipidemia (for definition, see Methods) was very high in both genders throughout the entire 30-year survey period with a significant decrease in both genders (males: from 87.5 to 74.8%; $p < 0.001$; females: from 87.7 to 69.9%; $p < 0.001$).

## Discussion

This has been the first study monitoring the trends in major CV risk factors in a representative sample of the Czech population over a period of more than 30 years, using standardized

methods originally developed by the WHO MONICA Project [13]. Similar longitudinal population data have been—to our knowledge—available only in Sweden [14], Lithuania [16] and Finland [15]. Continuous favorable changes in smoking habits in males, an overall significant decrease in BP and total cholesterol values, as well as a rise in the awareness, treatment and control of hypertension in both genders were found. However, within the same period, there was no change in smoking habits in females, whereas obesity increased significantly in males.

## Response rates

There was a significant downward linear trend in the response rates in both sexes with a sharp decline between the last two surveys. The overall response rate varied from 43.1 to 88.4%. The response rate in our last survey was 43.1% in males and 48.6% in females which may seem to be low, yet similar to other more recent or ongoing epidemiological studies in various parts of the world [17,18]. However, a possible selection bias due to the increased participation of healthier and more health-conscious individuals in the later surveys could not have been eliminated.

## Trends in cigarette smoking

The Czech MONICA and post-MONICA studies showed a decline in smoking prevalence only in males (from 45% in 1985 to 23.9% in 2016/17), and no change in females (varying from 20.9–25.9%). The latest, still very high smoking rates have substantially contributed to the continuously high cardiovascular morbidity and mortality in the Czech Republic. On the other hand, the US National Health Interview Survey reported only 15.5% of adults as current smokers (17.5% in males and 13.5% in females) in 2016. [19]. Worldwide in 2015, the age-standardized prevalence of daily smoking was 25.0% in men and 5.4% in women, representing 28.4% and 34.4% reductions, respectively, since 1990 [20]. Overall, 41.2% of CVDs were attributable to smoking in 2015.

In Europe, since the1980s, the prevalence of smoking in men has decreased in almost all European countries with available data, except for Latvia and Russia where the rates have risen to more than 50%. Within the same period, the prevalence of smoking in women has also decreased in most European countries but less than in males. The most recently available statistics on smoking in Europe reported higher overall prevalence in adult men (27.1%) than in adult women (18.5%) [7]. The prevalence of smoking in men was the highest in Eastern Europe and in the former Soviet Union countries. Contrarily, the smoking rates in women were very low in most former Soviet Union countries, except the Russian Federation (16.3%), however, Central and Eastern European countries displayed the rates comparable to Northern, Western, and Southern Europe, mostly around 20%.

## Trends in anthropometric parameters

A significant increase in body height and weight was observed in both genders of the representative Czech population sample over the entire study period. However, the weight gain was more pronounced in men (+10.4 vs + 3.0 kg), consequently increasing their body mass index (BMI) (from $27.0 \pm 4.0 \ kg/m^2$ in 1985 to $29.2 \pm 5.1 \ kg/m^2$ in 2016/17; $p<0.001$). Over the past century, adult height has changed variably across countries, experiencing a steady gain in most of them [21]. Changes in the population mean height did not correlate with the changes in mean BMI in men and a weak inverse correlation was found for women [22]. Consistent with the NCD Risk factor Collaboration analysis [21], the Czech men were ~12–13 cm taller than the Czech women; this difference remained constant over the period of 31/32 years.

There are several measures of body fatness available, however, most robust data refer to BMI. This index is used to define body weight categories [23]. Increased BMI is generally caused by long-term energy imbalance between calories consumed and calories expended. Over the past 50 years, obesity prevalence has increased worldwide, achieving pandemic dimensions [24]. Changes in the global food system together with an increased sedentary lifestyle seem to be the main driving forces of the obesity pandemic, with overnutrition becoming a greater health threat than undernutrition.

Czech consumption data have continuously shown a significant decrease in the consumption of meat (substantially reducing beef consumption and increasing consumption of poultry), sausage, eggs, and milk and dairy products since 1989. Animal dietary fats have been largely replaced by vegetable fat and oils. There has been also a significant increase in fresh fruit and vegetable consumption [25]. The food consumption data were supported by three studies based on the household budget survey conducted in 1991, 1994 and 1997, confirming positive changes in eating patterns [26].

Paradoxically, despite the favorable changes in food consumption and eating patterns in the Czech population after 1989, there was no change in BMI in women, and even a significant increase occurred in men (from $27.0 \pm 4.0$ kg/m$^2$ in 1985 to $29.2 \pm 5.1$ kg/m$^2$ in 2016/17; $p<0.001$) across all the age groups. Consequently, there was an immense increase in obesity in Czech males with its prevalence almost doubled (from 19.7% in 1985 to 37.7% in 2016/17; $p<0.001$). The proportion of overweight males also increased significantly, whereas in women the proportion of obese and overweight individuals remained more or less stable, nevertheless, high over the entire period (the overall prevalence of obesity in women ranging from 25.5 to 30.0%). The increase in obesity in the Czech males may have been partly attributed to quitting smoking, a pattern observed in many countries [27].

The NCD Risk Factor Collaboration investigators detected substantial regional differences in BMI changes over time, with a faster increase observed in South and Southeast Asia, the Caribbean and Southern America [22]. Similarly, there were substantial regional differences observed in the prevalence of obesity, varying from 3.7% in Japan to 38.2% in the USA [28].

A systematic review of representative studies published from 1990 to 2008, reporting obesity prevalence in Europe, had already identified the Czech Republic as one of the countries with the highest prevalence of obesity (i.e. >25%) [29].

Mean BMI increased in all European countries in men and almost in all of the women except for Belarus, Belgium, Bulgaria, Croatia, Czech Republic and Estonia [7]. The prevalence of obesity was higher in individuals with lower education, and with low physical activity. Overweight and obesity were more prevalent in males in the majority of European countries, including the Czech Republic.

## Trends in blood pressure, prevalence, awareness, treatment, and control of hypertension

A significant documented decline in the population mean SBP and DBP over a period of 31/32 years definitely contributed to a substantial decrease in CHD and stroke mortality in the Czech population. All BP changes were more pronounced in females and in age groups of 45–64 years. The population mean BP is generally affected by lifestyle changes and drug treatment of hypertension. As drug treatment of hypertension increased particularly in individuals over 45 years, it is likely that the BP changes in the age group 45–64 years were more closely related to drug treatment while the younger individuals were possibly more susceptive to lifestyle changes. In the Czech Republic, similarly to other high-income countries, the decline in BP was observed despite the increase in BMI, an otherwise established risk factor for hypertension

[22]. Other dietary factors related to lower BP include a higher intake of dietary fiber and potassium as well as a decrease in the consumption of saturated fats [30]. The Czech consumption data have documented a daily increase in the consumption of fruit and vegetables by ~30–35% since 1985 [25], which is—together with a reduced consumption of meat and sausages–an indirect evidence of reduced salt intake. On the other hand, these favorable BP changes might have been slowed down by the elevated alcohol consumption, with the Czech Republic belonging to the top three countries in the consumption of alcohol in Europe, and insufficient physical activity, reported by 28% of Czech males and 35% of Czech females in 2016 [1].

The overall prevalence of hypertension decreased only in females but not in males; a detailed analysis by age groups showed a significant decline in males in the age group 35–44 years and 45–54 years even though their BMI increased significantly. Nevertheless, the prevalence of hypertension in both genders remained very high. Our prevalence rates were in agreement with the findings of NCD Risk Factor Collaboration [31] and with the systematic analysis of population-based studies from 90 countries [32], indicating a persistently high prevalence of hypertension in Central and Eastern Europe.

Awareness, treatment, and control of hypertension improved significantly in both genders over a period of more than 30 years. As to the awareness and treatment of hypertension, our rates were slightly better than the overall data for high-income countries in 2010 [32], and much better than the USA data for the white population in 2011–2016 [19]. Thus the major problem in our population seems to be ineffective treatment of hypertension, resulting in poor control of hypertension (29.8% in men and 37.0% in women) despite the rise in combination therapy.

## Trends in lipid parameters

A significant decrease in total and non-HDL cholesterol together with the decrease of prevalence of dyslipidemia has been observed in a representative sample of the Czech population since 1985. The IMPACT model attributed the largest reduction in CHD death (39.5%) in the Czech Republic to the reduction in total cholesterol [9] which was predominantly induced by favorable dietary changes. The use of lipid-lowering drugs in the Czech general population was still rather low (14.6% in males and 10.0% in females) and did contribute to an improvement in lipid profile mostly in patients with manifest cardiovascular disease.

The decline in total cholesterol in most Western countries was the net effect of an increase in HDL-cholesterol and a decline in non-HDL-cholesterol [33]. Contrarily, our 30-year data showed just a small increase in HDL-cholesterol in younger females (aged 25–44 years) and even a small decrease in HDL-cholesterol in males. This may be due to the fact that healthy lifestyle, including recommended physical activity and refraining from smoking, has been mostly adopted by the younger female population.

## Strengths and limitations

All seven cross-sectional surveys were conducted in the same six districts of the Czech Republic, consistently respecting the seasonal variation. The methods used were standardized, originally developed by the WHO MONICA Project [13]. All lipid parameters were analyzed in the same lipid laboratory, formerly serving as the WHO reference laboratory for the MONICA Project. The blood pressure measurement technique did not change throughout the entire period, using a gold standard mercury sphygmomanometer. Moreover, the study period covered the transition from the totalitarian regime to democracy in the Czech Republic, associated with substantial socio-economic changes and their consequent impact.

A potential study limitation is the decline in the response rate which may have resulted in a population sample with different social characteristics, such as a higher level of education,

usually associated with a higher degree of health consciousness, thus making the results more favorable compared with the total general population. Nevertheless, most recent epidemiological studies worldwide have reported a response rate below 40%.

## Conclusions

There was a significant improvement in most major cardiovascular risk factors in the Czech population between 1985 and 2016/17. This period covered the transition from the totalitarian regime to democracy associated with socio-economic changes. Decreases in total cholesterol, the population mean BP and smoking rates in males most likely contributed to the significant decrease in cardiovascular mortality. On the other hand, there was no change in smoking habits in females, moreover, obesity increased substantially in males.

Our results, obtained from a representative population sample, have provided a valuable source of information about the population health, allowing for extrapolation to the entire Czech population. The analysis of longitudinal trends in CVD risk factors should help to analyze the trends in cardiovascular morbidity and mortality, which is of utmost importance for health care planning, particularly now, when there are data using the IMPACT model to explain the decline of CHD mortality in the Czech Republic providing evidence that more than 50% of the fall in CHD mortality have been attributable to the reduction in major CV risk factors [9].

## Supporting information

**S1 Data.**
(PDF)

**S1 File.**
(ZIP)

## Author Contributions

**Conceptualization:** Renata Cífková, Jan Bruthans, Peter Wohlfahrt, Jiří Widimský, Jr, Jan Filipovský, Otto Mayer, Jr, Zdenka Škodová, Věra Lánská.

**Data curation:** Renata Cífková, Peter Wohlfahrt, Alena Krajčoviechová, Jiří Widimský, Jr, Jan Filipovský, Otto Mayer, Jr, Zdenka Škodová, Petr Stávek, Věra Lánská.

**Formal analysis:** Renata Cífková, Peter Wohlfahrt, Alena Krajčoviechová, Jan Pudil, Aleš Linhart, Jiří Widimský, Jr, Jan Filipovský, Otto Mayer, Jr, Zdenka Škodová, Rudolf Poledne, Petr Stávek, Věra Lánská.

**Funding acquisition:** Renata Cífková, Jiří Widimský, Jr.

**Investigation:** Renata Cífková, Jan Bruthans, Peter Wohlfahrt, Alena Krajčoviechová, Pavel Šulc, Lenka Eremiášová, Jan Pudil, Jan Filipovský, Otto Mayer, Jr.

**Methodology:** Renata Cífková, Peter Wohlfahrt, Alena Krajčoviechová, Aleš Linhart, Jiří Widimský, Jr, Jan Filipovský, Otto Mayer, Jr, Zdenka Škodová, Petr Stávek, Věra Lánská.

**Project administration:** Renata Cífková, Jiří Widimský, Jr, Otto Mayer, Jr, Zdenka Škodová.

**Resources:** Renata Cífková.

**Software:** Peter Wohlfahrt, Alena Krajčoviechová, Aleš Linhart, Otto Mayer, Jr, Věra Lánská.

**Supervision:** Renata Cífková, Jan Bruthans, Peter Wohlfahrt, Alena Krajčoviechová, Pavel Šulc, Marie Jozífová, Lenka Eremiášová, Aleš Linhart, Jiří Widimský, Jr, Jan Filipovský, Otto Mayer, Jr.

**Validation:** Renata Cífková, Jan Bruthans, Peter Wohlfahrt, Alena Krajčoviechová, Pavel Šulc, Marie Jozífová, Lenka Eremiášová, Jiří Widimský, Jr, Otto Mayer, Jr, Rudolf Poledne, Petr Stávek, Věra Lánská.

**Writing – original draft:** Renata Cífková, Jan Bruthans, Peter Wohlfahrt, Alena Krajčoviechová, Pavel Šulc, Marie Jozífová, Lenka Eremiášová, Jan Pudil, Aleš Linhart, Jiří Widimský, Jr, Jan Filipovský, Otto Mayer, Jr, Zdenka Škodová, Rudolf Poledne, Petr Stávek, Věra Lánská.

**Writing – review & editing:** Renata Cífková, Jan Bruthans, Peter Wohlfahrt, Alena Krajčoviechová, Pavel Šulc, Marie Jozífová, Lenka Eremiášová, Jan Pudil, Aleš Linhart, Jiří Widimský, Jr, Jan Filipovský, Otto Mayer, Jr, Zdenka Škodová, Rudolf Poledne, Petr Stávek, Věra Lánská.

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
