## [Decision Letter · Decision Letter 0]

25 Mar 2020

PONE-D-20-04207

30-year trends in major cardiovascular risk factors in the Czech population, Czech MONICA and Czech post-MONICA, 1985 – 2016/17

PLOS ONE

Dear Dr. Cífková,

Thank you for submitting your manuscript to PLOS ONE. After careful consideration, we feel that it has merit but does not fully meet PLOS ONE’s publication criteria as it currently stands. Therefore, we invite you to submit a revised version of the manuscript that addresses the points raised during the review process.

We would appreciate receiving your revised manuscript by May 09 2020 11:59PM. To enhance the reproducibility of your results, we recommend that if applicable you deposit your laboratory protocols in protocols.io, where a protocol can be assigned its own identifier (DOI) such that it can be cited independently in the future. For instructions see: http://journals.plos.org/plosone/s/submission-guidelines#loc-laboratory-protocols

We look forward to receiving your revised manuscript.

Kind regards,

Tatsuo Shimosawa, M.D., Ph.D.

Academic Editor

PLOS ONE

Journal Requirements:

4. Please include your tables as part of your main manuscript and remove the individual files. Please note that supplementary tables (should remain/ be uploaded) as separate "supporting information" files

Reviewers' comments:

Reviewer's Responses to Questions

**Comments to the Author**

1. Is the manuscript technically sound, and do the data support the conclusions?

Reviewer #1: Yes

Reviewer #2: Yes

2. Has the statistical analysis been performed appropriately and rigorously? 

Reviewer #1: I Don't Know

Reviewer #2: Yes

3. Have the authors made all data underlying the findings in their manuscript fully available?

Reviewer #1: Yes

Reviewer #2: Yes

4. Is the manuscript presented in an intelligible fashion and written in standard English?

Reviewer #1: No

Reviewer #2: Yes

5. Review Comments to the Author

Reviewer #1: March 5, 2020

Re; PONE-D-20-04207

Dear editor,

Thank you for the opportunity to review the manuscript entitled, 30-year trends in major cardiovascular risk factors in the Czech population, Czech MONICA and Czech post-MONICA, 1985 – 2016/17, by authors Cífková et al.

I have no conflicts to report. I am not a population epidemiologist and do not have an extensive background in this type of data analysis.

I applaud the efforts of the authors for trying to analyze dynamic trends among the Czech population over the 30-year study period, amidst a huge change in politics and lifestyle. Population studies are fraught with many variables and the authors point out variability in response rates during the study period. Still there is value in reporting trends.

The POS ONE criteria for publication include:

The article is presented in an intelligible fashion and is written in standard English.

The content and use of verbal tense need to be reworked. The present tense should be avoided and replaced with the passive or past tense. (many sentences contain both the present and past tense.) I am not sure who is responsible for this type of editing. The discussion needs to be rewritten.

Typo’s:

Page 10: paragraph 2, line 4. Change ‘claims’ to ‘has claimed’

Page 10: paragraph 2, line 5. Change ‘is a growing’ to ‘has been a growing’

Page 16: paragraph 1, line 5. Change ‘one’ to ‘groups’

Page 16: paragraph 2, line 4. Change ‘one’ to ‘group’

Page 19: paragraph 2, line 10. Change ‘Fat of animal origin were’ to ‘Animal dietary fats were’

Page 20: first line is confusing and should be re-written.

Discussion needs to be rewritten.

Thus in its current form, I cannot recommend publication.

Reviewer #2: I found this article to be very informative and well written.

Minor comment

The graph shown in Figure 1E gives the impression of upward trend of HDL-C values in females, from 1992 to 2017. However, the temporary downward trend in 2001 (only in age groups 45-65 years) seems to be an unexpected observation. It would be useful to explain it.

6. PLOS authors have the option to publish the peer review history of their article (what does this mean?). If published, this will include your full peer review and any attached files.

Reviewer #1: No

Reviewer #2: No

---

## [Author Response · Author response to Decision Letter 0]

19 Apr 2020

Tatsuo Shimosawa, MD, PhD

Academic Editor

PLOS ONE

April 15, 2020

Dear Dr. Shimosawa,

Thank you very much for considering our manuscript for publication. We are very grateful to the reviewers for their valuable comments and suggestions. 

Please find below our response to the points raised.

Journal Requirements:

1. Style requirements have been adhered to.

2. The data set with all the individual data from 7 cross sectional surveys have been uploaded as a Supporting information file as well as a file entitled Data_coding_specification.

A link to access the data directly from the hospital server has been established at

http://www.ftn.cz/data-monica-1117/

3. The phrase “data not shown” has been deleted as source data have become available.

4. Tables 1 – 5 were included into the main manuscript.

Answers to the Reviewers’ comments”:

Reviewer #1: The comments were mostly related to style and use of English which has been taken into account and an appropriate revision of the text has been done. All the individual typo’s have been corrected. Substantial changes have been introduced in the Discussion section as indicated in the “Revised Manuscript with Track Changes”, pp 11-18. 

Reviewer #2: I found this article to be very informative and well written.

Minor comment

The graph shown in Figure 1E gives the impression of upward trend of HDL-C values in females, from 1992 to 2017. However, the temporary downward trend in 2001 (only in age groups 45-65 years) seems to be an unexpected observation. It would be useful to explain it.

We are very grateful for this comment which has lead us to check Table 5. The significance value for HDL-cholesterol for the age group 25-34 years has been, therefore, changed from non-significant to p<0.05 (there was a typing error in the previous Table 5 version). 

The text in the Results section (pages 10-11, lines….) has been changed as follows: 

Over the same period, there was also a significant but small decline in HDL-cholesterol in males (from 1.35 ± 0.36 to 1.34 ± 0.36 mmol/L; p <0.001) and no significant overall change in females (from 1.57 ± 0.36 to 1.71 ± 0.43 mmol/L; ns). Nevertheless, a significant increase in HDL-cholesterol was observed in the two youngest female age groups. Three-way ANOVA found All the main effects (sex, age groups, year of examination) were found significant (Fig. 1E; three-way ANOVA).

Appropriate modifications have also been done in the Discussion section: 

The decline in total cholesterol in most Western countries was the net effect of an increase in HDL-cholesterol and a decline in non-HDL-cholesterol [33]. However, Contrarily, our 30-year data did not showed just a small increase in HDL-cholesterol in younger females (aged 25-44 years) any HDL-cholesterol changes in females and even a small decrease in HDL-cholesterol in males. This may be due to the fact that healthy lifestyle, including recommended physical activity and refraining from smoking, has been mostly adopted by the younger female population.

Should there be any further questions, please do not hesitate to contact me.

Yours sincerely,

Renata Cífková

Center for Cardiovascular Prevention Charles University in Prague, First Faculty of Medicine and Thomayer Hospital

Vídeňská 800 

140 59 Prague 4

Czech Republic

---

## [Decision Letter · Decision Letter 1]

23 Apr 2020

30-year trends in major cardiovascular risk factors in the Czech population, Czech MONICA and Czech post-MONICA, 1985 – 2016/17

PONE-D-20-04207R1

Dear Dr. Cífková,

We are pleased to inform you that your manuscript has been judged scientifically suitable for publication and will be formally accepted for publication once it complies with all outstanding technical requirements.

With kind regards,

Tatsuo Shimosawa, M.D., Ph.D.

Academic Editor

PLOS ONE

Additional Editor Comments (optional):

Reviewers' comments:

Reviewer's Responses to Questions

**Comments to the Author**

1. If the authors have adequately addressed your comments raised in a previous round of review and you feel that this manuscript is now acceptable for publication, you may indicate that here to bypass the “Comments to the Author” section, enter your conflict of interest statement in the “Confidential to Editor” section, and submit your "Accept" recommendation.

Reviewer #1: All comments have been addressed

Reviewer #2: All comments have been addressed

2. Is the manuscript technically sound, and do the data support the conclusions?

Reviewer #1: Yes

Reviewer #2: Yes

3. Has the statistical analysis been performed appropriately and rigorously? 

Reviewer #1: I Don't Know

Reviewer #2: Yes

4. Have the authors made all data underlying the findings in their manuscript fully available?

Reviewer #1: Yes

Reviewer #2: Yes

5. Is the manuscript presented in an intelligible fashion and written in standard English?

Reviewer #1: Yes

Reviewer #2: Yes

6. Review Comments to the Author

Reviewer #1: The authors have corrected grammatical errors.

Their questionnaire consisted of "yes/no' answers and thus they did not quantify data i.e., smokers (cig/day). I think they could have made more powerful statements about diet/smoking habits as they related to political changes. None the less they have pointed out interesting trends in the Czech population over time.

Reviewer #2: (No Response)

7. PLOS authors have the option to publish the peer review history of their article (what does this mean?). If published, this will include your full peer review and any attached files.

Reviewer #1: No

Reviewer #2: No

---

## [Editor Report · Acceptance letter]

28 Apr 2020

PONE-D-20-04207R1 

30-year trends in major cardiovascular risk factors in the Czech population, Czech MONICA and Czech post-MONICA, 1985 – 2016/17 

Dear Dr. Cífková:

I am pleased to inform you that your manuscript has been deemed suitable for publication in PLOS ONE. Congratulations! Your manuscript is now with our production department. 

With kind regards,

on behalf of

Prof. Tatsuo Shimosawa 

Academic Editor

PLOS ONE